# Human Resource Management in Sustainable Development

Viera Sukalova [1], Zuzana Stofkova [1] and Jana Stofkova [2,*]

1    Department of Economics, University of Zilina, Univerzitna 1, 010 26 Zilina, Slovakia
2    Department of Communication, University of Zilina, Univerzitna 1, 010 26 Zilina, Slovakia
*    Correspondence: jana.stofkova@fpedas.uniza.sk; Tel.: +241-0415133201

**Abstract:** The article deals with human resource management and selected personnel indicators in sustainable development. The main goal of this paper is to analyse and describe human resource management, focused on the audit of workload, the motivation of employees, the competence of staff, their knowledge and experience. The article shows how the selected personnel variables are investigated in selected establishments of a multinational company in Slovakia, as an attractive employer interested in the growth of its employees, thus employer branding. For the research methods, a case study methodology is used. Data collection was carried out through a study of the employees' performance and through a survey with employees in the company, as well as an interview with the sale advisors and managers. The research aimed to point out a case study of the investigation of the selected personnel indicators in human resources management in a selected company and to examine the perception of the performance of the employees in connection with their financial evaluation and their satisfaction within the selected company and the impact on the employees' performance. Furthermore, the research aimed to determine whether there was a dependence between the subjectively perceived performance of the employees and the selected aspects using the statistical SPSS program. Further indicators were calculated, such as workload, proportion of wasted time and labour productivity.

**Keywords:** human resource management; personnel audit; variables; COVID-19 pandemic; performance of employees; motivation

## 1. Introduction

Nowadays, the negative trends in the economies of some countries are forcing organisations to look for ways to increase the effectiveness of human resource engagement [1]. The global financial and economic crisis, and the COVID-19 pandemic has intensified the human capital's focus and its effective management. The author of [2] sees human resources as the most important assets and valuable resources in terms of business performance. According to the authors [3], they define the issue of human resource management as a philosophy that creates an incalculable value for the company. Human resource practices are considered valuable because they strengthen the company's performance. The analysis of [4], emphasizes the areas which are key to the development of the organisation, human capital, measurement and audit, internal and external communication, infrastructure and technology. With the rise of globalization, it is necessary to define the concept of international human resource management. It represents the broader issue of human resource management focused on multinational companies and the management of cross-cultural work procedures, with an orientation towards the development and maintenance of the functions and processes associated with the issue of human resource management [5]. The authors of [6] claim that the following human resources management activities are the most commonly practiced: the creation of work systems, the recruitment of employees, the education of employees, replacements and compensations and the implementation of new activities. Companies which realise the efficient use of human resources can benefit their

company in the form of cost reduction and increased profits. For this reason, when selecting potential employees, companies are increasingly focusing on people's experience and competencies, in light of the company's goals and objectives [7]. Thanks to their knowledge, experience and skills, the company's market value increases. In general, the term 'personnel policy' is associated with personnel strategy, which defines the strategic and corporate goals, that the company wants to pursue in the future [8]. The employees, who are also sufficiently motivated and rewarded, represent a competitive advantage for the company [9]. By obtaining comprehensive information about the atmosphere among the employees, the process reserves and performance capacities, as well as the risks, the negatives and weaknesses of the company is realized by personnel controlling (Figure 1) [10].

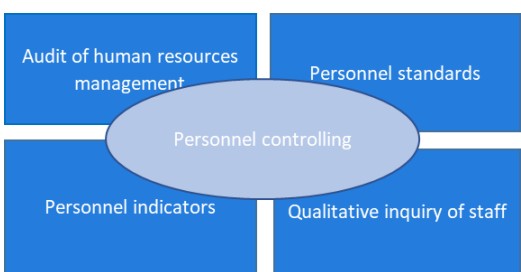

**Figure 1.** Tools of personnel controlling, processed according to source Controller Institut, Prague, 2004 in [10].

Personnel audit—a strategic tool of personnel controlling is to provide feedback and at the same time, this tool serves to evaluate the current state of the human resources management in the company. With the help of a human resources audit, we obtain a broader control of the results, as well as the efficiency and effectiveness of personnel management. Examples of the focus of a human resources audit are: the search for opportunities to optimize the process of recruiting and selecting employees, the audit of employee performance evaluation and the audit of the current evaluation system [11].

Audits, from the focus point of view, are carried out in practice at different levels. In companies, we may encounter, for example, financial, managerial or personnel audits. [12]. In practice, we encounter different types of audits, not all of which are mandatory for an organisation. Some audits are legally required, e.g. accounting audits. There are non-mandatory audits that are carried out on the initiative of the organisation, such as a personnel audit [13]. We recognise also, the audit of employees' satisfaction, the audit of the use of the working time fund, the audit of staff stability and turnover, the audit of interpersonal communication, the audit of the strengths and weaknesses of individual workers, etc. [14]. The HR audit is a tool through which we can independently assess the quality of human resources, the level of personnel management and the effectiveness of the organisational structure of the company. The HR audit can result in specific recommendations concerning key employees of the company, such as managers, for example. Based on the results of the audit, the company is able to make changes at the organisational level [15].

Personnel indicators—the aggregate variables that express the relationship between several variables. A prerequisite for effective HRM indicators is the establishment of the criteria (e.g. optimum level, height, standard) against which their fulfilment can be assessed. Auditing is the collection and evaluation of evidence to determine and report on the degree of correspondence between the information and the established criteria [16]. Thanks to the personnel audit, the management of the company obtains information on the basis of which the company can manage its activities more effectively and also work more efficiently with the available human resources [17].

The personnel audit assesses the company's personnel policy. It collects and evaluates the information on whether the company's personnel policy is set correctly and effectively. It is based on monitoring the procedures and the specific activities of the company. These

procedures should be in accordance with the specific plan of the company, but also within its corporate culture [8].

In stage 0, the decision to conduct a personnel audit must be based on the set goals. The preparatory phase involves the collection of data, based on an interview, questionnaires, observations, self-assessment, etc. The proposals for the company are processed through an evaluation audit report, the measures and calculation of the costs associated with the implementation of the measures. The last phase is the implementation of the measure into practice. Based on the results of the audit, the company can make important decisions. It is necessary to repeat the audits at regular intervals, approximately once every two years. Such recurring audits provide feedback for the company regarding the benefits from the introduction of measures. (Figure 2) [18].

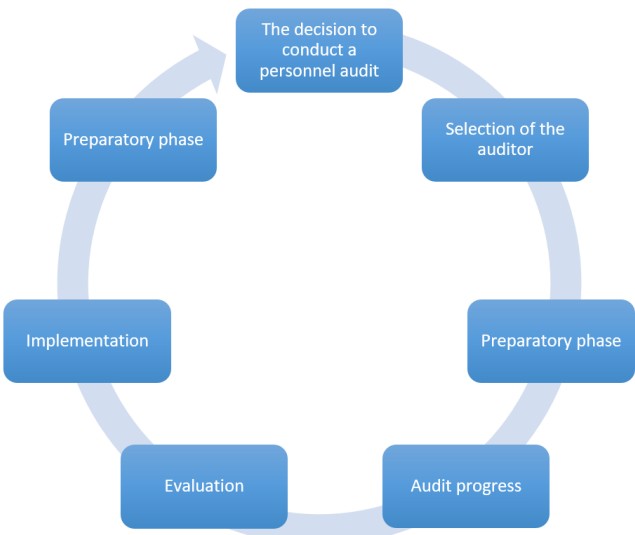

**Figure 2.** Audit process. (own processing).

One of the decisive prerequisites [19–21] for the effective fulfilment of the objectives of an organization's development strategy, is the quality and optimal structure of the human resources. The starting point for HR is the relevant and objective information obtained through the HR audits. This can be carried out continuously [22–25].

The proper and efficient functioning of a company is largely influenced by its employees. There is no single definition of the term personnel audit, but many opinions agree that a personnel audit is a human resource management tool. A personnel audit provides a company with the valuable formation needed to improve the management and assess the company's personnel situation. The continuous improvement of intra-company employee relations can help a company to remain competitive or reduce the amount of employee turnover in the company. Personnel auditing is an important activity in the management of a company. Although it is a costly and time-consuming process, it has its justification in every company, regardless of its size and structure. It is important to recognize that the money spent on personnel audits is ultimately of great value to the company [26–29].

To be effective in the long term, professionals must have an integrative vision and comprehensive skills across the functional as well as the corporate plans that enable the product to move quickly along the production channel, and to reach the end point—the customer—as quickly as possible. The ability to harness the power of HR policies is necessary, in order to ensure that the HR programs are effectively executed and enforced throughout the company, throughout the supply network [30–33]. In order to be in demand in the professional sphere, a person must continuously deepen his knowledge and skills. It is very important for companies to pay attention to the education and development of their employees, because this is the only way for the company to achieve sustainable development [34].

Hajdu and Sebestyén identified different factors leading to employee satisfaction, motivation for a better performance or reward system. They consider human resources as a key element of business development that changes the competitiveness of enterprises. In their work, they stated that an effective reward system increases the loyalty and quality of employees' performance and consequently increases the competitiveness of firms. When employees' needs are accurately identified, personal development occurs, resulting in an increased efficiency and a better performance [35].

Employee motivation is also discussed in [35], who examine its impact on employee performance. The analysis focuses on the characteristics of the workers and their impact on the results achieved in the organization. In their study, they distinguish between the concept of motivation, which they define as the intrinsic energy and personal will to act, and work motivation, which is understood as a measure of an employee's self-motivation to perform effectively. The process of strategic change, which is important to the organization, is also contingent on the roles of managers and leaders and their coordination. Managers manage the resources of the organization, including human resources, and through this coordination, they perform various functions or tasks.

Mulyani, in their study, defined work motivation as the willingness of an employee to exert great effort to achieve the goals of the organization, conditioned by the ability to exert effort to satisfy some individual needs. The purpose of their study was to find out the effect of leadership and reward on motivation and also its implications on the performance of cooperative employees [36].

In order to be in demand in the professional sphere, a person must continuously deepen his knowledge and skills. It is very important for companies to pay attention to the education and development of their employees, because this is the only way for the company to achieve sustainable development [37].

According to a 2020 study, training is seen by employees as one of the main factors that increase their motivation to perform at work. Motivation consequently acts as a catalyst for the employee to complete the assigned tasks better than the normal routine [38].

## 2. Materials and Methods

The selected company operates in the field of the manufacturing industry and it is a multinational company and it is expected to grow in the future. Currently, the multinational company owns 5076 establishments in 65 countries. The survey was conducted using a quantitative method, in the form of a survey using Google Forms, is furthered by observations, and expert interviews with human resources managers. The employees represented the target group, involving 64 sale advisors and first level managers. In our survey, we collected data from employees from two operations of the company. Considering the fact that the compliance of respondents during the COVID-19 pandemic was lower in this period, and many employees had to be released because of a lower turnover and fewer clients. The survey aimed to analyse the human resource management focused on the process of human resource management and audit the competence of staff, and the actual use of their knowledge and experience in the company, as well as the sufficient motivation of employees. The level of reliability was determined at 95%, with a standard deviation 0.5, and a confidence interval of 10%. The number of employees in the two selected establishments reached 64.

The survey was performed from March to April 2021. The electronic questionnaire consisted of 18 questions, a verbal scale (Likert scale), and open items. The obtained data were processed using the automatic processing of Google Forms and Microsoft Excel. We monitored the dependence of the ordinal variables and the binary variables. The tools of descriptive statistics were used. To verify the hypotheses, the "Fisher test" and the "Chi-squared test" were used to calculate the test statistics.

The size of the research sample was determined by the exact statistical calculation quantified, based on the relationship:

$$n = \frac{Z^2 * (p) * (1 - p)}{c^2} n = \frac{1.96^2 * 0.5 * (1 - 0.5)}{0.05^2} = 384.16 \, respondents \qquad (1)$$

$$\text{New } n = n \, / (1 + (n - 1) / pop) n \cdot n_{kor} = \frac{384.16}{1 + \frac{384.16 - 1}{64}} = 54.89 \, respondents$$

where $n$ represents the minimum number of respondents, $p$—the percentage of respondents who know or do not know the issue—the maximum error specified by us (p-admissible gauge margin 0.5%), $z$—the critical value of the normal distribution at a significance level $\alpha = 0.05$ (95% estimation reliability).

The respondent size sample was according to equation 54, the level of significance was set to be 0.10, which corresponds to a 90% confidence interval. Therefore, a representative sample should be 54 respondents, where pop represents the population size. The questionnaire was sent to 64 managers. A total of 54 valid questionnaires were returned, which meant a return of 84.4%.

In terms of the age structure of the respondents, the largest group of respondents fell in the age category of 22 to 25 years, with 36%. The second largest group fell in the age category from 26 to 29 years, with 24%. The rest of the respondents (24%) were between the ages of 18 to 21. The majority of respondents were sales advisors, department managers, visual managers and store managers. In the survey, the sample consisted of 13% men and 87% women. The company mostly employs people with a completed secondary education with a high school diploma—up to 67%. Seventeen percent of employees have obtained a first-level university degree and 7% of employees have a second-level university degree.

Of the interviewed employees, 63% have been employed by the company for one to two years. Twenty-four percent of employees have been working for the company for less than one year. Only 2% of employees can be considered long-term, as they have been working for the company for more than 5 years.

To test the hypotheses, the SPSS program was used. We used Fisher's exact test, because it was not possible to use the Chi-squared test for the dependencies due to the low frequency of some answers. In evaluating this test, we set a significance level of $\alpha = 0.05$.

## 3. Results

The aim of the survey was to obtain the data from the employees' perspective on the company. We focused on information regarding their satisfaction with the company in terms of workload, benefits and motivation from the employer. Fifty-four employees were questioned as part of the survey and the return rate of the questionnaires was 84.4%. The survey focused on the employees' opinion of the company.

Up to 63% of employees are of the opinion that the amount of assigned work corresponds to their salary. Seventeen percent of employees feel that they sometimes do not have enough work and therefore do not use their full potential. Conversely, 15% of employees feel a heavy work load on certain days or hours. Only 5% of employees feel constantly overworked and believe that an additional employee is needed.

As part of the survey, we found out whether employees feel that their work allows them to use their abilities, education and skills. Eighty-seven percent of employees answered positively and only 7% of employees disagreed with this statement.

Furthermore, we investigated the most common reasons for the inability of employees to fully utilize their potential. Most employees said that the lack of customers prevents them from doing so. Another influencing factor was the lack of experience and insufficient motivation on the part of the employer. Based on these results, we can say that it is necessary to ensure sufficient knowledge of work procedures and increase motivation on the part of the employer (Figure 3).

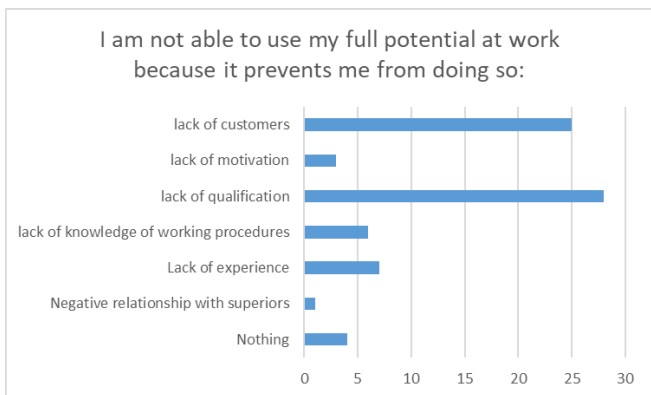

**Figure 3.** The use of employees' potential.

The company places great emphasis on regular feedback from superiors, which we can also notice, based on the development of the images of employees' working days. Eighty-seven percent of respondents said that feedback from their supervisor is also important for them, so that they can continuously improve their work. Considering the company's policy and their emphasis on following established procedures, we can say that the company has its tactics set correctly. As many as 93% of employees answered that they receive regular feedback from their superiors.

The financial evaluation of employees is one of the most important motivating factors for employees to perform. Fifty percent of employees are completely satisfied with their financial compensation and up to 35% of employees are rather satisfied. A total of 15% of employees show dissatisfaction with their financial compensation (Figure 4).

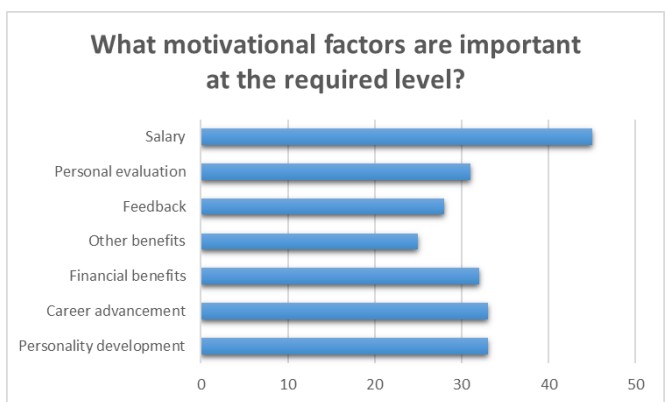

**Figure 4.** Employees' motivational factors.

Furthermore, we investigated which motivational factors influence the work performance of employees the most. Based on the obtained data, we can claim that the most motivating factor is salary. Moreover, the company's employees consider their personal growth, their personal evaluation and the possibility of their career growth as further motivational factors as well.

According to the survey, the respondents were provided with sufficient information about the employees' opinion of their work. The majority of employees completely agreed with the opinion that their work team is: energetic, communicative, well-coordinated and professional. To be familiar with the company's visions and goals is very important for the company. For this reason, new employees have an obligation to familiarize themselves with the company's goals and must put in practice these goals and visions. According to the survey, 59% of employees claim that they know the goals and visions of the company. On the contrary, up to 41% of employees do not have a sufficient knowledge about this issue.

It is also important that employees are identified with the goals and visions of the company. Otherwise, we can observe a negative impact on employees in the performance of their work. According to the survey, 48% of employees did not identify with the company's visions and goals, which represents a very significant result that needs to be explored in depth. Such a result represents the possibility that employees do not share the corporate culture of the company, which can affect the effectiveness of their work.

The relationship with the company is an important factor affecting employee satisfaction. Thirty-five percent of employees are proud to work for this company. Thirty percent of employees are happy to have at least some work and 26% have a neutral attitude towards the company. In the results, 9% of employees admitted that they do not wish to talk about working for this company.

The company provides its employees with various financial and other benefits as a form of employee motivation. Other benefits include regular team building activities for employees and organised by the company. An important motivating factor is also the improvement of qualifications through internal employee training. Financial benefits, in practice, have a much greater impact on employee satisfaction, which is why the company mainly provides employees with meal tickets and allowances for recreation after two years of employment. Apart from the statutory contributions, the company offers employees the possibility of a contribution to their supplementary pension insurance and a holiday contribution, in the case of employees who have worked for the company for at least 2 years.

The career growth is a factor that affects the number of long-term employees in the company. If employees feel that they have nowhere to move, their productivity and intrinsic motivation will gradually decrease. However, up to 62% of respondents are of the opinion that the company allows them to grow their career within the company. Twenty-fiver percent of respondents could not answer this question and 13% of employees think that the company does not provide them with such an option.

The recommendation of the company is a positive sign for the company. However, such a recommendation will often provide companies with the right people for various job opportunities. Up to 71% of respondents would recommend the company as an employer to other people. Twelve percent of respondents said that they would not recommend the company as an employer.

Based on the respondents' answers, we can claim that the company does not perform regular personnel audits, within the company. As many as 59% of respondents confidently claim that they have not been part of a personnel audit in the past. Although 41% of the respondents could not answer this question, based on these answers we can assume that the personnel audit in the company did not take place during the duration of their employment relationship.

The following statements from the inquiry were analysed:

- Workload of the employees during the day, for the respondents.
- Share of unnecessary loss of time, caused by the employee.
- Share of unnecessary loss of time, caused by technical and organisational deficiencies.
- Share of the possible increase in labor productivity after the removal of unnecessary time consumption, caused by the employee.
- Share of the possible increase in work productivity after the elimination of unnecessary time consumption, caused by technical and organisational deficiencies.
- The total percentage of the possibility of increasing labor productivity.

The following table shows the calculations of the individual indicators, on the basis of which we can evaluate the use of the shift time of the selected employee (Table 1).

**Table 1.** Selected personnel indicators in an employee´s performance.

| Variable | Sample | Calculation | Result |
|---|---|---|---|
| Workload of the employee (K1) | $x = \frac{T1+T2}{T} \times 100$ | $x = \frac{T1+T2}{T} \times 100$ | 96.27% |
| Proportion of time wasted, caused by an employee (K2) | $x = \frac{T2'-T2+TD}{T} \times 100$ | $x = \frac{49-30+0}{510} \times 100$ | 3.73% |
| Proportion of time wasted caused by the technical and organisational shortcomings (K3) | $= \frac{TE}{T} \times 100$ | $= \frac{19}{510} \times 100$ | 3.73% |
| The share of the possible increase in labor productivity from the elimination of unnecessary time consumption, caused by the employee (K4) | $x = \frac{T2'-T2+TD}{T-(T2'-T2+TD+TE)} \times 100$ | $x = \frac{49-30+0}{510-(49-30+0+19)} \times 100$ | 4.03% |
| Share of the possible increase in labor productivity after the elimination of unnecessary time consumption, caused by the technical and organisational deficiencies (K5) | $x = \frac{TE}{T-(T2'-T2+TD+TE)} \times 100$ | $x = \frac{19}{510-(49-30+0+19)} \times 100$ | 4.03% |
| Total percentage of the potential for productivity improvement (K6) | K4 + K5 | 4.03 + 4.03 | 8.05% |

The table shows the calculations of the individual indicators that allow us to evaluate the use of shift time of a selected employee. The workload of this employee reached 96.27%, which was a positive result. The proportion of unnecessary time lost on behalf of the employee and the employer, is the same—it represents 3.73% of the working time. Following the elimination of the errors that cause an unnecessary loss of time, it is possible to increase the employee's productivity by 4.03%. In total, we can increase labor productivity by up to 8.05%. The survey shows that the majority of employees were satisfied with their work in the company. The majority of employees consider their workload to be average, which is related to the fact that they are satisfied with the financial evaluation of their work. The biggest motivation for employees was their salary, possible career growth and personal evaluation. The biggest shortcomings were observed in the employees' identification with the company's visions and goals. The company attaches a great importance to the fact that employees know and identified with the company´s culture, but not all employees are familiar with the company´s visions and goals.

*Workday Snapshot Evaluation*

The company set a weekly work pool of 40 hours, which varies according to the employee's working hours. We evaluated the snapshots of the working days of the managers. The aim was to have an overview of the employees' work routines in order to analyse thoroughly analyse their workload and, if necessary, to define the losses that can be eliminated and thus make the employees' working time more efficient. Based on the data collected on the employee's working activities, we then tabulated these times and expressed the individual work time and rest time as a percentage. We then compared this data with the standardized times in order to be able to compare the efficiency of the employee's work.

The survey also showed that employees were not able to use their skills and experience fully, mainly due to lack of motivation of the procedures and a lack of customers. All of these reasons are obstacles for the employer, and for this reason it is necessary to create an environment that will allow employees to develop and improve their skills. We evaluated the dependencies of the individual answers of the respondents.

In the frame of the research, the questions were set:

- Is there a statistically significant difference between the workload and the length of employment of the employee?
- Is there a statistically significant difference between the length of employment and receiving feedback from the chief?

- Is there a statistically significant difference between the completed educational activities and the financial evaluation of employees?
- Is there a statistically significant difference between the workload and employees' financial compensation?

The first hypothesis (H1) was set as follows:

**H$_0$.** *The workload and the length of employment variables are independent variables.*

**H$_1$.** *The workload and the length of employment variables are dependent variables.*

An important indicator in this test is the "Exact Sig. (2—sided)", based on which, we accept or exclude the H$_0$ or H$_1$ hypothesis. The value of the indicator is lower than the significance level $\alpha$ and therefore we reject hypothesis H$_0$, which means that we consider the selected variables to be dependent. We can conclude that the employees gain experience during the years of employment and by attending the training programs in the company (Table 2).

**Table 2.** Chi-squared test.

| | Value | df | Asymptotic Significance (2sided) | Exact Sig. (2sided) |
|---|---|---|---|---|
| **Chi-Squared Tests** | | | | |
| Pearson Chi-Squared | 52.718 [a] | 9 | 0.000 | 0.000 |
| Likelihood Ratio | 42.685 | 9 | 0.000 | 0.000 |
| Fisher–Freeman–Halton Exact Test | 34.164 | | | 0.000 |
| N of Valid Cases | 54 | | | |

[a] Fourteen cells (87.5%) have an expected count of less than five. The minimum expected count is 11.

Next, we used Fisher's test to investigate whether the length of the employment relationship influences the receipt of feedback from supervisors. We set hypothesis (H2):

**H$_0$.** *The length of employment and receiving feedback variables are independent.*

**H$_1$.** *The length of employment and receiving feedback variables are dependent variables.*

Based on the exact significance, we found out that this variable is higher than the significance level of 0.05 and hence, we accepted hypothesis H$_0$, that the variables are independent. In practice, the length of employment should not affect the regularity of the feedback to employees. Giving feedback is considered as an effective tool to motivate employees. The chi-squared test results show that we accept hypothesis H$_0$, and thus the workload and financial evaluation variables are independent. This result is significant for the company and signals the need for corrective action in the area of employee financial compensation. The employees should be evaluated according to their performance. The evaluation, according to determined variables, is greatly recommended. If employees are underpaid and there is too much pressure on their workload, employee dissatisfaction may increase rapidly over time.

Further, we investigated the dependence of employees' financial evaluation on their education. We set hypothesis (H3):

**H$_0$.** *The education and financial evaluation of employees variables are independent.*

**H$_1$.** *The education and financial evaluation of employees variables are dependent.*

The exact significance is higher than the significance level, so we accept hypothesis H$_0$. The financial evaluation variable is not dependent on education. Considering the

interview with the managers working for the company, we confirmed through testing, the dependency of the responses that the education of the employees is not very important for the performance of the work at each store. Much more relevant, are their personal motivations and work experience. For this reason, the financial evaluation is also not based on the highest educational attainment of the employee.

Based on the survey, we can claim that employees are most affected by the lack of customers, insufficient experience and insufficient motivation, on the part of the employer. Next, we examined the impact of the workload on employees' financial evaluation, which can largely affect employee satisfaction and negatively affect their motivation. In the case of a high employee workload and an insufficient financial remuneration, employee dissatisfaction is very likely to arise, which may lead to the employee´s resignation or burnout syndrome. For these two variables, we set hypothesis (H4):

**H$_0$.** *The workload and the employee's financial compensation variables are independent.*

**H$_1$.** *The workload and the employee's financial compensation variables are dependent.*

The chi-squared test results show that we accept the hypothesis H$_0$, and thus the workload and financial evaluation variables are independent. This result is significant for the company and signals the need for corrective action in the area of employee financial compensation. If employees are underpaid and there is too much pressure on their workload, employee dissatisfaction may increase rapidly over time (Table 3).

**Table 3.** Chi-squared test.

| Chi-Squared Tests | | | | |
|---|---|---|---|---|
| | **Value** | **df** | **Asymptotic Significance (2sided)** | **Exact Sig. (2-sided)** |
| Pearson Chi-Squared | 86.839 [a] | 9 | 0.000 | 0.000 |
| Likelihood Ratio | 65.782 | 9 | 0.000 | 0.000 |
| Fisher–Freeman–Halton Exact Test | 53.597 | | | 0.012 |
| N of Valid Cases | 54 | | | |

[a] Fourteen cells (87.5%) have an expected count of less than five. The minimum expected count is 11.

Based on the results of the survey, we found that not all employees received regular feedback from their supervisors. The information from this communication process can then be used effectively to monitor the progressive improvement of the employee's competencies, and can also be used in the employee's career development. In spite of these opportunities, we encounter situations where employees are not made aware of such opportunities. For this reason, we propose a system of regular feedback from the supervisor. Employee interviews should be held every three months in order to ensure that each employee who shows potential for career development or increased effectiveness in his or her work, is treated individually. The survey shows that staff feel insufficiently motivated in their work. Although the company provides employees with the opportunity for career development, this form of motivation is insufficient for many employees. However, there are no rewards for employees when they meet their sales targets. Hence, we propose a financial reward in the form of a percentage reward when the set plan is met. Such a reward positively influences the behavior of employees who feel a sense of being sufficiently rewarded and are able to perform better in their job. Based on the results of the survey, we found that a number of employees are not sufficiently familiar with the working procedures used in the company. One of the reasons why this is the case, is considered to be the lack of staff training and regular awareness of these procedures. The goal is to improve the regular training process for employees. At the same time as increasing their competencies, the selected employee may perceive such a situation as a form of career growth and positive motivation.

## 4. Discussion

The effective functioning of society is influenced to a great extent by the employees of the company. The definition of the term 'personnel audit' is not clear, although according to many opinions, a personnel audit represents a toll on human resources management. Based on the personnel audit, the company gains valuable data, which is important for the more effective management and evaluation of the personal indicators of the company. Improving intra-company relations with employees can maintain the competitiveness of the company or decrease the fluctuation rate of the employees.

According to [15], a personnel audit has two key forms, namely a human resources audit and an audit of the organization. An audit of the HR management system focuses on the tools, procedures, competencies and effectiveness of the HR management, inter alia, in connection with the certification of its functions Moreover, the main objective of the organizational audit is to optimize the organizational structure and the personnel systematization of the organization.

Through a personnel audit, a company can obtain answers regarding the functioning of the essential processes in the organization, such as: fulfilling the organization's vision, meeting business and economic indicators, meeting quality indicators and customer requirements, functionality and efficiency of the organizational structure, the effective functioning of teams, sufficient motivation of employees, ascertaining their opinions and attitudes, the competence of staff and the actual use of their knowledge and experience, corporate culture and the implementation of change [20].

Based on the information about the company, it is necessary that the auditor is able to use the appropriate methods to obtain the necessary data. The aim of the survey was to conduct a personnel and organizational audit and based on the results, to develop proposals and measures for the selected company. The topic of the personnel and organizational audits and its importance in practice, is undisputable. We carried out a personnel and organizational audit in a specific company, based on the selected indicators which were investigated in the research.

One group of theorists and practitioners considers personnel auditing as a function of personnel marketing and personnel management. The other group is inclined to understand a personnel audit as a comprehensive method or tool for obtaining relatively accurate and specific data and information on the current state of human capital in a company [20].

As mentioned above, based on the personnel audit, we can reveal the strengths and weaknesses of the employees, their potential and abilities to perform the work entrusted to them. On the basis of this information, we can analyse the current state of employees and therefore identify their shortages or surpluses. From each analysis, the company can plan the next steps that need to be initiated. Among such steps, we can include, for example, the possibility of additional training of employees, change in their competencies, structural and organizational changes of the company, etc. Such changes can bring several benefits to the company, such as cost reduction, reduction of employee turnover or increase in employee commitment through appropriately chosen motivations [21].

Management requires answers to several questions in the personnel audit:

Is the company's management (processes, structure, leadership) efficient and effective? Is the company's management sufficiently informed about the management of the company? Are all of the resources available to the company being used efficiently? Based on these questions, the company is able to search for new opportunities for company development and innovations that lead to the improvement of the company's functioning process [22,23].

The American approach considers the HR audit as an important aspect of corporate image building, while quantifying the social consequences of the organization's activities.

The French approach conceives of the personnel audit as a tool for the mutual understanding between management and the workers of the company. The combined approach is a combination of the American and French approaches. The main role of this approach is to align the interests of the organization and the interests of the employees [24].

At the moment when the company is not sure whether they have the right number of people with the right experience and skills, or some employees are overloaded, it is advisable to carry out a desk audit.

The audit should verify that the company's policies, procedures and that the documents relating to recruitment, retention, discipline, termination and postemployment are fair and legal [25].

A staff audit is a type of functional audit. A human resources audit consists of diagnosing, analyzing, evaluating and assessing future courses of action within the company. The personnel audit is an essential tool for the management of the company. Its objective is not only to obtain an audit and quantification of results, but also to adopt a broader perspective that will be helpful in defining the future directions of the human resource activities [18].

The HR audit is an important tool for determining the effectiveness of the HR policy implemented in an organization. The audit monitors and controls the procedures and activities taking place in the company so that these activities are in line with the corporate culture and the organization's plan [8].

According to [19], a personnel audit is "an effective tool that assesses, in depth, the strengths and weaknesses of employees, their potential with respect to their current job position, while helping to create the right structure of teams."

According to [26], through a personnel audit, it is possible to reveal the objective information, on the basis of which it is then possible to work more effectively with the company's employees. The audit complements the management's knowledge of the quality of their human resources with an external assessment that is objective and independent.

According to [27], a HR audit also provides an opportunity to assess the financial advantages and disadvantages of the HR functions, compare these functions, evaluate the function effectiveness, ensure compliance, establish standards, bring HR closer to the managers and improve the quality of employees. In addition, based on the results of the HR audit, it is possible to determine the areas of the company's human resources function that have the greatest potential for a return on investment.

According to [28], "The main objective of the personnel audit is a comprehensive, correct and professional reassessment of factors that subjectively or objectively affect the efficiency and performance of the company or its individual components."

According to [15], the author characterizes the personnel audit as a means of independent assessment of the quality of the company's employees, the effectiveness of the company's organizational structure, as well as assessing the level of the personnel management. The results of such an analysis are the recommendations that are linked to the development of the company. They mostly concern the use of key employees of the company, the improvement of management efficiency and the overall improvement of the organizational structure of the organization.

According to [29], a personnel audit is a good way to assess an organization's performance. Based on the data obtained, it is possible to identify activities that are inefficient or performed in a duplicated manner, the suitability of the requirements for each position, the workload of employees, as well as their personality profile, with regard to their strengths and weaknesses. Nowadays organizations decide how and by what methods to assess staff, especially their professional competencies, which are an important determinant of their future career development. Personnel audits have an important predictive capacity for the company's needs, the company can take measures that can ensure the growth of the company and the achievement of its objectives at a minimum cost.

We can assess the interdependence between the organizational structure and managerial effectiveness indicators through indicators, such as enterprise innovation, productivity, legitimacy, employee loyalty and the quality of employee performance [26].

Development, change in the organizational structure and increasing the efficiency of work in the company are the main pillars of the personnel audit. Based on the results of the personnel audit, the company can create a procedure, thanks to which it can reduce the time required to hire new employees. For this reason, it is necessary for the company to

determine the key processes of the company [30]. The development of society, the reduction of unemployment and thus the economic development can be achieved through quality, well-functioning and financially secured education system [31,32].

Nowadays, it is very difficult to survive in the competitive environment and it is necessary to use all opportunities to reduce the costs and increase profits or opportunity for company development. The dynamic nature of today's business environment forces managers to regularly monitor and evaluate the performance of the company. The supply chain management focuses on the management of the supply chain, thereby significantly affecting the performance of the company.

Finally, the results are calculated accurately using software to determine the deviation between the optimal solution and the output of the algorithms [38].

As well as the NLP model of the inventories in multi-level supply chains (SCs). The presented NSM can give a direct method with a predictable level of fill without pivoting, decreasing the number of taken iterations to find the optimum solution. We transform the nonlinear equations into an NLP model, which the NSM can solved [39].

The supply chain managers and investors can determine the optimal strategies for competing in the market by observing the different parameters, such as risk-taking, expected profit and product type. As the retailer is one of the crucial components of an SC, their decision can directly affect the manufacturers' pricing strategies. The managers of the high quality product manufacturer should be aware that if retailers decide to increase the quantity of an order, they will have to raise capital through the platform to compete, which will increase interest rates. The increase in costs will lower their profit margins and make it difficult to compete.

The high quality product manufacturer is suggested to invest in economies that have good market potentials for those products [40].

On paper, they design and optimise an integrated four-level supply chain (SC), which contains a supplier, a producer, a wholesaler and multiple retailers. The levels cooperate to make an integrated SC (ISC) so that the inventory cost is minimised and the reliability is maximised, simultaneously [41].

Accordingly, a closed-loop supply chain (CLSC) with multi-stage products is designed with respect to the green production principles and the quality control (QC) policy under back-logged and lost sale types of the shortage of supplies. An integrated reliable five-level closed-loop supply chain with multi-stage products under quality control and green policies is a generalised outer approximation with an exact penalty.

Alireza Amjadian & Abolfazl Gharaei (2022), in their work, design and optimise an integrated five-level supply chain (SC), which contains a supplier, a producer, a wholesaler, multiple retailers, and a collector. The levels cooperate with each other to make an integrated supply chain (ISC) so that the total cost function is minimised and the total reliability function is maximised, simultaneously [42].

The goal is to minimise the cost function and maximise the profit function under stochastic constraints, simultaneously [43–46].

## 5. Conclusions

A personnel audit is a highly effective tool for the company for its development and advancement. Within the audit of personnel indicators, we carried out the survey about the employees' opinions in the multinational company.

We obtained feedback in the form of representative information on the needs and attitudes of employees, on the management of human resources in two establishments of a multinational company, as well as on the problematic areas and the need for more attention in the future, and to take the necessary measures to improve them. The aim of the survey was to obtain the data from the employees' point of view. However, its aim was also to distinguish the personnel variables from the perspective of the company. The inquiry focused on the satisfaction with the company in terms of workload, benefits and motivation from the employer. Based on the data obtained, we were able to calculate the

individual indicators of lost working time and the possibilities of increasing the efficiency of the employees, by eliminating the detected errors. The selected indicators, such as the workload, the proportion of time wasted caused by an employee, the proportion of time wasted, caused by technical and organizational shortcomings, the share of the possible increase in labor productivity from the elimination of unnecessary time consumption, caused by the employee, or by the technical and organizational deficiencies, the total percentage of the potential for productivity improvement in selected employees who showed satisfactory levels, which can be monitored and improved. The implementation of the personnel indicators secures a low fluctuation of the employees and a sustainable development of the company presented in the higher performance of the employees.

The evaluation of the employees, according to the determined variables is greatly recommended. According to the survey, 59% of employees claim that they knew the goals and visions of the company and 48% of employees did not identify with the company's visions and goals, which represents a very significant result that needs to be explored in depth. We recommended a regular assessment of the employees every three months to give them feedback about their workload, satisfaction and motivation. We recommended that the company perform regular personnel audits within the company.

**Author Contributions:** Conceptualization, V.S.; methodology, Z.S.; software, V.S.; validation, J.S.; formal analysis, V.S.; investigation, J.S.; resources, V.S.; data curation, Z.S.; writing—original draft preparation, V.S.; writing—review and editing, Z.S.; visualization, Z.S.; supervision, V.S.; project administration, J.S; funding acquisition, J.S. All authors have read and agreed to the published version of the manuscript.

**Funding:** This research was funded by VEGA 1/0460/22.

**Institutional Review Board Statement:** Not applicable.

**Informed Consent Statement:** Not applicable.

**Data Availability Statement:** Not applicable.

**Conflicts of Interest:** The authors declare no conflict of interest.

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
