# Peer review of "Human Resource Management in Sustainable Development"

_sustainability, doi:10.3390/su142114258_

Round 1

Reviewer 1 Report

The topic discussed in the light of the covid-19 pandemic is interesting, but its implementation requires a thorough change. The current shape of the article does not correspond to the scientific discussion.

- First of all, the received article is a draft version, as evidenced by unremoved changes, and red selections.

- In the article, it is necessary to make an in-depth review of the scientific literature, which would indicate the research carried out in a given field and the contribution of the authors.

- Numerous hypotheses are put forward in the article, but they have not been proven.

- The issue of factors affecting motivation has not been exhausted.

- The conclusions need to be supplemented.

Author Response

-The topic discussed in the light of the covid-19 pandemic is interesting, but its implementation requires a thorough change. The current shape of the article does not correspond to the scientific discussion.

Dear reviewer, thank you very much. We are very grateful for your kind and polite appreciation of our work. Thank you for your arduous work. We would like to express our great appreciation to you for your comments on our paper.  

- First of all, the received article is a draft version, as evidenced by unremoved changes, and red selections.

Thank you very much for your comment. We have made revisions regarding draft version by unremoved changes, and red selections.

- In the article, it is necessary to make an in-depth review of the scientific literature, which would indicate the research carried out in a given field and the contribution of the authors.

Thank you very much for recommendation to add the scientific literature. We added these references the scientific literature as you are recommended:

- Numerous hypotheses are put forward in the article, but they have not been proven.

H0: The variables length of employment and getting feedback are independent.

H1: The variables length of employment and getting feedback are dependent variables.

This result is significant for the company and signals the need for corrective action in the area of employee financial compensation. The employees should be evaluated according to their performance. The evaluation according to determined variables is greatly recommended.

The issue of factors affecting motivation has not been exhausted.

In order to be in demand in the professional sphere, a person must continuously deepen his knowledge and skills. It is very important for companies to pay attention to the education and development of their employees, because only in this way can the company achieve sustainable development (Váchal et al. 2013, p. 688).

Hajdu and Sebestyén (2021) identified different factors leading to employee satisfaction, motivation for better performance or reward system. They consider human resources as a key element of business development that changes the competitiveness of enterprises. In their work, they stated that an effective reward system increases the loyalty and quality of employee performance and consequently increases the competitiveness of firms. When employees' needs are accurately identified, personal development occurs, resulting in increased efficiency and better performance.

Smutny (2020) in his paper "The impact of manager's personality on employee motivation" tried to highlight the main roles of a manager in a company, focusing on his emotional and social intelligence, which form an important part of his personality and can bring positive changes on employee motivation

- The conclusions need to be supplemented.

Thank you very much for your comment. We have made revisions regarding the conclusion.

Reviewer 2 Report

I have reviewed the paper, titled “Human resource management and selected personnel indicators in sustainable development".

Comments:

A)     Relevance: The paper is within the areas of interest of Sustainability journal.

B)    Title: The paper title reflects the main core of the provided novelties. However, the title can be shortened and condensed.

C)   Keywords: The provided keywords are not enough.

D)   Originality and contribution: The paper conveys some novelties and contributions. It seems to be eligible for publications after doing the minor modifications as follows:

E)    The communication aspect of paper suffers from some grammatical mistakes and typos. I recommend to revising some main sections, including Abstract and Conclusion with the help of a native editor.

F)    I had a deep review on research literature. I found that the enough attention has not been paid to describe the literature and the related issues to Reference section. It is so necessary to have a comprehensive review on references not only for providing a correct form of citations, but also for detail description of new published papers.

G)   The introduction and literature section should be improved by the top papers in the fields of supply chain (SC), because of the importance of human resource management in SCs. It is heavily recommended to enriching mentioned sections by adding the recent published papers as follow. The author(s) don't need to describe the below papers in detail manner, but they are heavily expected to mention the name of author(s) and publication year for more reviews of readers. You can open a parenthesis and write: for deep reviews, consider the below references.

*(doi.org/10.1007/978-3-030-89743-7_10) *(doi.org/10.1080/23302674.2022.2083254) *(doi.org/10.1007/s10479-022-04648-w)  *(doi.org/10.1080/23302674.2021.1958023) *(doi.org/10.1080/23302674.2021.1919336) *(doi.org/10.1080/23302674.2021.2015007)

H)   It is strongly recommended to add ALL OF the above references not only in the manuscript, but also at the end of paper (Reference section) in a detail manner, accurately.

I)      Please delete old references. Maintain references from 2010 to 2021.

J)     Assumptions: I believe that this paper is configured based on some certain assumptions. Assumption(s) may be obvious, but they must be clearly stated.

K)    Applications: Who would benefit from the paper and how? How valuable your results are to managers? What suggestions you have for them?

L)    I recommend authors to going in deep in managerial implications. Therefore, a stronger effort in reporting recommendations is heavily requested.

M)   Limitations and Future research: What are the other limitations of this research? So, what do author(s) suggest as future research in order to cover the limitations. It is strongly recommended to mentioning aforesaid issues in “Conclusion” section.

N)   At first, the above minor modifications should be done, accurately. Then, the paper must be reviewed, again. I will make my decision, after doing the above modifications in a detail manner, accurately.

Author Response

I have reviewed the paper, titled “Human resource management and selected personnel indicators in sustainable development".

Comments:

  1. Relevance: The paper is within the areas of interest of Sustainability

Dear reviewer, thank you very much. We are very grateful for your kind and polite appreciation of our work. Thank you for your arduous work. We would like to express our great appreciation to you for your comments on our paper.  

  1. B)    Title: The paper title reflects the main core of the provided novelties. However, the title can be shortened and condensed.

             Thank you for your arduous work and instructive advice. We tried to implement it.

  1. C)   Keywords: The provided keywords are not enough. Thank you for your instructive advice. We tried to implement it.

  1. D)   Originality and contribution: The paper conveys some novelties and contributions. It seems to be eligible for publications after doing the minor modifications as follows:

      Dear reviewer, thank you very much for your comment. We have made revisions which are marked in yellow colour.

  1. E)    The communication aspect of paper suffers from some grammatical mistakes and typos. I recommend to revising some main sections, including Abstract and Conclusion with the help of a native editor.

Thank you very much for your comment. We would like to undergo the revision by the native-speaker editor.

  1. F)    I had a deep review on research literature. I found that the enough attention has not been paid to describe the literature and the related issues to Reference section. It is so necessary to have a comprehensive review on references not only for providing a correct form of citations, but also for detail description of new published papers.

Dear reviewer, thank you very much for your comment.  We have made revisions in the article regarding the literature and the related issues added to Reference section as you proposed which are marked in yellow colour.

  1. G)   The introduction and literature section should be improved by the top papers in the fields of supply chain (SC), because of the importance of human resource management in SCs. It is heavily recommended to enriching mentioned sections by adding the recent published papers as follow. The author(s) don't need to describe the below papers in detail manner, but they are heavily expected to mention the name of author(s) and publication year for more reviews of readers. You can open a parenthesis and write: for deep reviews, consider the below references.

Finally, the results are calculated accurately by GAMS software to determine the deviation between the optimal solution and the output of the algorithms. A Multi-product EPQ Model for Defective Production and Inspection with Single Machine, and Operational Constraints: Stochastic Programming Approach Reza Askari, Mohammad Vahid Sebt & Alireza Amjadian

*(doi.org/10.1007/978-3-030-89743-7_10)

In this paper, an NSM optimises the constrained NLP model of the inventories in multi-level supply chains (SCs)..  An integrated lot-sizing policy for the inventory management of constrained multi-level supply chains: null-space method Abolfazl Gharaei,Amir Amjadian,Alireza Amjadian,Ali Shavandi,Ahmad Hashemi,Mahdi Taher & show all, Received 11 Feb 2021, Accepted 23 May 2022, Published online: 09 Jun 2022

*(doi.org/10.1080/23302674.2022.2083254)

The supply chain managers and investors can determine optimal strategies for competing in the market by observing different parameters such as risk-taking, expected profit and product type. As the retailer is one of the crucial components of an SC, their decision can directly affect the manufacturers' pricing strategies.

Online peer-to-peer lending platform and supply chain finance decisions and strategies

Ata Allah Taleizadeh, Aria Zaker Safaei, Arijit Bhattacharya & Alireza Amjadian

Annals of Operations Research volume 315, pages397–427 (2022) *(doi.org/10.1007/s10479-022-04648-w)  

In this paper, we design and optimise an integrated four-level Supply Chain (SC), which contains a supplier, a producer, a wholesaler and multiple retailers. The levels cooperate to make an Integrated SC (ISC) so that the inventory cost is minimised and the reliability is maximised, simultaneously.

An integrated reliable four-level supply chain with multi-stage products under shortage and stochastic constraints, Abolfazl Gharaei,Alireza Amjadian &Ali Shavandi

Received 15 Mar 2021, Accepted 15 Jul 2021, Published online: 11 Aug 2021*(doi.org/10.1080/23302674.2021.1958023)

Alireza Amjadian & Abolfazl Gharaei (2022) in their work  design and optimise an integrated five-level Supply Chain (SC), which contains a supplier, a producer, a wholesaler, multiple retailers, and a collector. Accordingly, a Closed-loop Supply Chain (CLSC) with multi-stage products is designed with respect to the green production principles and Quality Control (QC) policy under back-logged and lost sale types of the shortage. Levels cooperate with each other to make an Integrated Supply Chain (ISC) so that the total cost function is minimised and the total reliability function is maximised, simultaneously

ALIREZA AMJADIAN & ABOLFAZL GHARAEI (2022) An integrated reliable five-level closed-loop supply chain with multi-stage products under quality control and green policies: generalised outer approximation with exact penalty, International Journal of Systems Science: Operations & Logistics, 9:3, 429-449, DOI: 10.1080/23302674.2021.1919336 *(doi.org/10.1080/23302674.2021.1919336)

The goal is to minimise the cost function and maximise the profit function under stochastic constraints simultaneously.

Abolfazl Gharaei,Seyed Ashkan Hoseini Shekarabi &Mostafa Karimi

Received 21 Aug 2021, Accepted 01 Dec 2021, Published online: 23 Dec 2021*(doi.org/10.1080/23302674.2021.2015007)

Dear reviewer, thank you very much for recommendation of valuable references. We added these references as you recommended:

  1. H)   It is strongly recommended to add ALL OF the above references not only in the manuscript, but also at the end of paper (Reference section) in a detail manner, accurately.

             The references we have corrected according to your recommendations. Further insight could      be drawn from:

  1. I)     Please delete old references. Maintain references from 2010 to 2021.

       Thank you for your instructive advice. We tried to implement the contemporary references from 2010-2021.

  1. J)    Assumptions: I believe that this paper is configured based on some certain assumptions. Assumption(s) may be obvious, but they must be clearly stated.

  1. K)   Applications: Who would benefit from the paper and how? How valuable your results are to managers? What suggestions you have for them?

   It is necessary to know the supply change management for the managers from the perspective of the future bussines and the profit of the company. For the managers the results are valuable above all from the perspective of building loyal and sustainable development within the organisation. From the perspective of the personnel department it is useful to select personnel indicators which are evaluated on a regular basis and to conclude principles of correct and fair evaluation which serves for fostering motivation in employees who work for the organisation with enthusiasm and adher to organisation´s principles.

      We would like to show with our research results that only employees thet are sufficiently motivated and rewarded are an competetive advantage of the company. Only what it is measured can be objectively appreciated in the terms of financial and non-financial evaluation and motivation of the emplyees.

  1. L)    I recommend authors to going in deep in managerial implications. Therefore, a stronger effort in reporting recommendations is heavily requested.

       Implementation of personnel indicators secures low fluctuation of the emplyoees and sustainable development of the company presented in higher performance of the employees. The issue of managerial implications were implemented in the text according to your recommendations. We thank you very much for your kind advice.

The results of the sensitivity analysis provide some meaningful insights to the practitioners (e.g. supply chain managers), stakeholders and decision-makers. The insights facilitate them in adopting the right strategies under disparate situations. The supply chain managers and investors can determine optimal strategies for competing in the market by observing different parameters such as risk-taking, expected profit and product type. As the retailer is one of the crucial components of an SC, their decision can directly affect the manufacturers' pricing strategies.

  1. M)   Limitations and Future research: What are the other limitations of this research? So, what do author(s) suggest as future research in order to cover the limitations. It is strongly recommended to mentioning aforesaid issues in “Conclusion” section.

We are grateful for your inspirative and polite recommendation. The limitation of our research resides in  the size of the sample and evaluation of further personnel indicators which we would like to investigate in our further research concerning an analysis of the correspondence of functional duties and the results of their professional activity, salary level, bonus policy. 

  1. N)   At first, the above minor modifications should be done, accurately. Then, the paper must be reviewed, again. I will make my decision, after doing the above modifications in a detail manner, accurately.

We tried to do our best according to your kind recommendations.

review 3:

Personnel audit based on a sociological survey. Accordingly, with the help of personnel audit, the revealed result is one-sided non-objective information. Likewise, key employees cannot be identify based on their own opinions. For such conclusions, there should be an analysis of the correspondence of functional duties and the results of their professional activity, salary level, bonus policy etc.

The research is interesting, but one-sided.

We are grateful for your inspirative and polite recommendation. We thank you for your work. We agree with the opinion of subjective evaluation of working attitudes and professional activity. We would like to address to these issues in our further research concerning an analysis of the correspondence of functional duties and the results of their professional activity, salary level, bonus policy.

We agree with the opinion of one-sided research because missing the objective issues in more places that it is presented. We tried to show it in the table 1 regarding the calculation concerning the personnel indicators, such as workload, prop

Reviewer 3 Report

Personnel audit based on a sociological survey. Accordingly, with the help of personnel audit, the revealed result is one-sided non-objective information. Likewise, key employees cannot be identify based on their own opinions.

For such conclusions, there should be an analysis of the correspondence of functional duties and the results of their professional activity, salary level, bonus policy etc.

The research is interesting, but one-sided

Author Response

Personnel audit based on a sociological survey. Accordingly, with the help of personnel audit, the revealed result is one-sided non-objective information. Likewise, key employees cannot be identify based on their own opinions. For such conclusions, there should be an analysis of the correspondence of functional duties and the results of their professional activity, salary level, bonus policy etc.

The research is interesting, but one-sided.

We are grateful for your inspirative and polite recommendation. We thank you for your work. We agree with the opinion of subjective evaluation of working attitudes and professional activity. We would like to address to these issues in our further research concerning an analysis of the correspondence of functional duties and the results of their professional activity, salary level, bonus policy.

We agree with the opinion of one-sided research because missing the objective issues in more places that it is presented. We tried to show it in the table 1 regarding the calculation concerning the personnel indicators, such as workload, proportion of wasted time and labour productivity, etc.